# MaMe: Matrix-Based Token Merging

## Abstract

We introduce MaMe, a training-free, differentiable token merging method that relies entirely on matrix operations to accelerate vision transformers. When applied to pre-trained models, MaMe doubles ViT-B@224 throughput with a mere 2% drop in accuracy. For training from scratch, a ViT-T model with MaMe achieves 1.94x throughput with a 1.3% accuracy drop. As a downsampling layer in Swin architectures, MaMe reduces FLOPs by 2.4x for Swin-S backbones, achieving 35.8% mIoU on ADE20K semantic segmentation. In SigLIP2-B@512 zero-shot classification, MaMe provides 1.3× acceleration with negligible performance degradation (78.02 vs. 78.37). For multimodal reasoning, MaMe accelerates LLaVA-v1.5-7B inference by 36% on MME with minimal degradation (31.40 vs. 32.76). In video tasks, MaMe accelerates VideoMAE-L by 48.5% on Kinetics-400 with a 0.84% accuracy loss. Collectively, these results demonstrate MaMe's effectiveness in accelerating transformer-based vision and multimodal models.

## 1 Introduction

Vision Transformers (ViTs)(Dosovitskiy et al., 2021) have revolutionized computer vision by adopting the transformer architecture from natural language models(Vaswani et al., 2017). However, the complexity of self-attention is quadratic $\mathcal{O}(N^2)$, where N represents the number of tokens. For applications requiring dense token representations, such as high-resolution images, this quadratic complexity presents a significant challenge, limiting the deployment of large-scale ViT models on resource-limited devices or in real-time applications.

To address the $\mathcal{O}(N^2)$ computational challenge, a straightforward yet effective approach is to reduce the number of tokens $N$ involved in the process. The strategies that have emerged include token pruning, token merging, and hybrid methods that integrate both. Pioneering works like DynamicViT(Rao et al., 2021) introduced a dynamic token sparsification framework that uses a lightweight, learnable prediction module to hierarchically prune tokens at various stages of the network. EViT(Liang et al., 2022) uses the class token to evaluate token importance, keeping the most attentive tokens while merging the others. Pruning's main drawback is irreversible information loss. Token merging combines similar tokens instead of discarding them. ToMe(Bolya et al., 2022) introduced a training-free method, using a fast bipartite soft matching algorithm to progressively merge similar tokens. Token Pooling(Marin et al., 2021) uses cluster analysis to aggregate information from neighboring tokens. DiffRate(Chen et al., 2023) makes compression rate differentiable to learn layer-wise rates, while Token Transforming(Zeng et al., 2025) generalizes both pruning and merging as specific cases of a broader matrix transformation, enabling more flexible, many-to-many mappings that can better preserve information.

Despite recent advancements, existing token reduction methods face several challenges. A primary issue is the **non-differentiable** nature of the token selection process when using the Top-K operation, which often requires complex workarounds for end-to-end training. Some methods are slow due to their reliance on clustering techniques like k-means, which are **computationally intensive** in practice. Additionally, many methods introduce **extra learnable parameters** for token selection or merging modules, leading to increased model complexity and training overhead. Lastly, an issue is the **dependency on specific architectures**; for example, EViT's reliance on a class token restricting its use in models where a class token might not be available.

To simultaneously address these limitations, inspired by ToMe, we introduce a training-free token merging approach that overcomes the mentioned challenges through several ways:

**Differentiable Design**: Our method employs only differentiable operations throughout the token merging process, enabling seamless end-to-end training. By avoiding discrete operations, we maintain gradient flow and allow the model to be trained from scratch.

**Efficient Matrix Operations**: Instead of relying on operations such as clustering algorithms, sorting or explicit maximum selection,we utilize efficient, GPU-friendly full-matrix operations. This approach offers both theoretical efficiency and practical speedup.

**Parameter-Free Architecture**: Our approach introduces no additional learnable parameters, maintaining the original model's parameter, simplifying deployment, and reducing the complexity of model management.

**Plug-and-Play Integration**: Our approach can be directly applied to pre-trained models without any extra training, or seamlessly integrated during training from scratch. This flexibility significantly lowers the barrier to adoption.

## 2 RELATED WORK

### 2.0.1 TOKEN PRUNING

Pruning methods discard non-informative tokens based on importance metrics. DynamicViT(Rao et al., 2021) pioneered this by using lightweight prediction heads to score token relevance, enabling end-to-end training. EViT(Liang et al., 2022) enhanced this by fusing pruned tokens into the class token while reducing sequence length. AdaViT(Meng et al., 2022) extends pruning to attention heads and transformer blocks, creating instance-adaptive computation graphs for complex inputs. However, these methods face limitations: 1) Early pruning risks information loss, 2) Discrete selection creates optimization challenges, and 3) Task-specific tuning is needed for threshold calibration.

### 2.0.2 TOKEN MERGING

Merging techniques combine similar tokens rather than discarding them, preserving information while reducing computational load. ToMe(Bolya et al., 2022) revolutionized this area with training-free bipartite soft matching to merge the most similar token pairs at each layer. However, ToMe's fixed merge ratio per layer limits adaptability to varying input complexities. DiffRate(Chen et al., 2023) addresses the challenge of selecting an optimal merge ratio by rendering the rate itself differentiable. It utilizes a learnable budget controller to optimize this rate for each input, facilitating instance-adaptive efficiency through standard gradient descent but increasing complexity. ToFu(Song et al., 2024) diverges from ToMe's training-free methodology by proposing a learnable fusion module that is co-trained with the models to generate new, more expressive tokens. Hybrid approaches such as Pumer(Fu et al., 2024) and LTPM(Li et al., 2024) integrate token pruning and merging within a unified framework. Pumer introduces a learnable router to dynamically determine the number of tokens to prune and merge on a per-instance basis, whereas LTPM employs learnable parameters to decide whether a token should be pruned or which tokens should be merged.

### 2.0.3 CLUSTERING-BASED REDUCTION

Clustering approaches use offline algorithms to group similar tokens. TCFormer(Zeng et al., 2024) employs KNN-enhanced Density Peaks Clustering to group tokens and merge redundant ones through averaging for human activity tasks like pose estimation. ClusTR(Xie et al., 2022) uses hierarchical token merging with cosine similarity across Transformer layers for vision tasks, but its fixed ratios limit flexibility and may hinder small object detection. While these methods preserve global context, they face three drawbacks: 1) Iterative clustering algorithms with O(nk) complexity offset computational gains, 2) Discrete cluster assignments prevent gradient flow, and 3) Fixed cluster counts lack input adaptability.

### 2.0.4 LEARNABLE TOKEN REDUCTION

End-to-end trainable methods optimize reduction policies through differentiable architectures. ATS(Fayyaz et al., 2022) implements token merging via weighted averaging with gating mechanisms. Dynamic Token Morphing(Wang et al., 2023) uses cross-attention between original and

learnable proxy tokens for information absorption. Gumbel Token Selector(Kim et al., 2023) employs Gumbel-Softmax to sample token subsets through residual connections. These approaches show promise but increase model complexity (15-30% more parameters) and risk overfitting on small datasets.

## 3 METHODOLOGY

**Token Partitioning** Let the input sequence from a given layer be represented by the matrix $X \in \mathbb{R}^{L \times d}$, where $L$ is the number of tokens and $d$ is the feature dimension. We first partition this sequence into two disjoint sets: a set of $M$ destination tokens, denoted by $\mathbf{X}_{\text{dst}} \in \mathbb{R}^{M \times d}$, and a set of $N$ source tokens, $\mathbf{X}_{\text{src}} \in \mathbb{R}^{N \times d}$, where $L = M + N$.

$$\mathbf{X}_{\text{dst}} = \{\mathbf{x}_i : i \in \mathcal{I}_{\text{dst}}\}$$
$$\mathbf{X}_{\text{src}} = \{\mathbf{x}_j : j \in \mathcal{I}_{\text{src}}\} \tag{1}$$

where $\mathcal{I}_{dst}$ and $\mathcal{I}_{src}$ represent the index sets for destination and source tokens, respectively, such that $\mathcal{I}_{dst} \cap \mathcal{I}_{src} = \emptyset$ and $\mathcal{I}_{dst} \cup \mathcal{I}_{src} = \mathcal{I}$ covers all token indices, excluding any special tokens (e.g., class tokens). The specific strategy for partitioning into $\mathcal{I}_{\text{dst}}$ and $\mathcal{I}_{\text{src}}$ can vary (e.g., alternating or random selection).

**Similarity Matrix.** We begin by computing the cosine similarity between each destination token and every source token. This yields a similarity matrix $S \in \mathbb{R}^{M \times N}$, where each element $S_{ij}$ is defined as:

$$S_{ij} = \frac{\mathbf{x}_i^{\text{dst}} \cdot \mathbf{x}_j^{\text{src}}}{\|\mathbf{x}_i^{\text{dst}}\| \cdot \|\mathbf{x}_j^{\text{src}}\|} \tag{2}$$

To isolate the most significant relationships, we apply a rectified linear unit (ReLU) activation with a shifting threshold $\tau$. This step filters out weak connections, producing a sparse similarity matrix $\tilde{S} \in \mathbb{R}^{M \times N}$:

$$\tilde{S}_{ij} = \text{ReLU}(S_{ij} - \tau) \tag{3}$$

**Adaptive Weight Pruning.** From the sparse similarity matrix $\tilde{S}$, we first compute an initial weight matrix $W \in \mathbb{R}^{M \times N}$ by normalizing its columns. This ensures the initial influence of each source token is properly distributed among its similar destination tokens.

$$W_{ij} = \frac{\tilde{S}_{ij}}{\sum_{i=1}^{M} \tilde{S}_{ij} + \epsilon} \tag{4}$$

where $\epsilon$ is a small constant for numerical stability.

To further refine these weights, we introduce a dynamic, column-specific thresholding mechanism. For each source token $j$, we define a threshold $\zeta_j$ as the average of its non-zero weights in $W$:

$$\zeta_j = \frac{\sum_{i=1}^{M} W_{ij}}{C_j + \epsilon} \tag{5}$$

where $C_j$ is the count of non-zero entries along the destination dimension and can be computed as

$$C_j = \sum_{i=1}^{M} \frac{W_{ij}}{W_{ij} + \epsilon} \tag{6}$$

For differentiability, we don't apply boolean operations $C_j = \sum_{i=1}^{M} W_{ij} > 0$.

The threshold $\zeta_j$ is to prune connections that are weak relative to a source token's other connections. We apply this threshold to obtain a pruned weight matrix $\tilde{W}$:

$$\tilde{W}_{ij} = \text{ReLU}(W_{ij} - \zeta_j) \tag{7}$$

Finally, the pruned matrix $\tilde{W}$ is re-normalized column-wise to produce the final fusion weights $W^{\mathrm{F}} \in \mathbb{R}^{M \times N}$:

$$W_{ij}^{\mathrm{F}} = \frac{\tilde{W}_{ij}}{\sum_{i=1}^{M} \tilde{W}_{ij} + \epsilon} \tag{8}$$

**Token Aggregation and Preservation** The destination tokens are updated by aggregating the features from source tokens, guided by the final fusion weights. The fused destination tokens, $X_{\mathrm{dst}}' \in \mathbb{R}^{M \times d}$, are computed as:

$$\mathbf{X}_{\mathrm{dst}}' = \mathbf{X}_{\mathrm{dst}} + \mathbf{W}^{\mathrm{F}} \mathbf{X}_{\mathrm{src}}$$

$$\mathbf{x}_{\mathrm{dst},i}'' = \frac{\mathbf{x}_{\mathrm{dst},i}'}{1 + \sum_{j=1}^{N} W_{ij}^{\mathrm{F}}} \tag{9}$$

A key component of our methodology is the preservation of unique source tokens $\mathbf{X}_{\mathrm{pres}}$. A source token $x_j^{\mathrm{src}}$ is preserved if it exhibits no similarity to any destination token, which means the sum of its similarities to all $M$ destination tokens is zero: $m_j = \mathbb{I}(\sum_{i=1}^{M} W_{ij}^{F} = 0)$, where $\mathbb{I}(\cdot)$ is the indicator function. So $\mathbf{X}_{\mathrm{pres}} = \{\mathbf{x}_j^{src} \mid m_j = 1\}$.

The final, reduced sequence is formed by concatenating any special tokens $\mathbf{X}_{\mathrm{spec}}$, the merged destination tokens $\mathbf{X}_{\mathrm{dst}}''$, and the set of preserved source tokens $\mathbf{X}_{\mathrm{pres}}$.

$$\mathbf{X}_{\mathrm{final}} = \mathrm{concat}(\mathbf{X}_{\mathrm{spec}}, \mathbf{X}_{\mathrm{dst}}'', \mathbf{X}_{\mathrm{pres}}) \tag{10}$$

**Batch Processing Implementation.** For efficient implementation on batched data, the preservation decision must be consistent across all samples in a batch. Given the fusion matrix for a batch be $\mathbf{W}^{F} \in \mathbb{R}^{B \times M \times N}$. A per-sample preservation mask $m^{(b)} \in \{0, 1\}^{N}$ is computed for each sample $b$, where $m_j^{(b)} = \mathbb{I}(\sum_{i=1}^{M} W_{bij}^{F} = 0)$. To ensure batch consistency, a source token $j$ is preserved if it is marked for preservation in *any* sample, yielding a final batch-wide mask $m_j^{\mathrm{final}} = \bigvee_{b=1}^{B} m_j^{(b)}$. So the preserved source tokens $\mathbf{X}_{\mathrm{pres}} = \{\mathbf{x}_j^{src} \mid m_j^{\mathrm{final}} = 1\}$. Subsequently, to prevent preserved tokens from fusing, the corrected fusion matrix $\tilde{\mathbf{W}}^{F}$ is obtained by $\tilde{\mathbf{W}}_{b,i,j}^{F} = \mathbf{W}_{b,i,j}^{F} \cdot (1 - m_j^{\mathrm{final}})$, zeroing out columns corresponding to preserved tokens and keeping others unchanged.

**Computational Efficiency** Given that $M$ and $N$ are fractions of the original sequence length $L$ (i.e., $M \approx \alpha L$, $N \approx (1 - \alpha)L$), this overhead scales approximately as $O(\alpha(1 - \alpha)L^2 d)$, similar to self-attention. When merging is applied, the subsequent attention computation is reduced to $O(L'^2 d)$. Assuming $L' \approx \beta L$ with $\alpha \leq \beta \leq 1$, this becomes $O(\beta^2 L^2 d)$. Therefore, the total cost is $O\left((\alpha(1 - \alpha) + \beta^2) L^2 d\right)$. For more efficient than standard self-attention, it requires $\alpha(1 - \alpha) + \beta^2 < 1$, which simplifies to the condition $\beta < \sqrt{\alpha^2 - \alpha + 1}$. To assess the strictness of this condition, we consider the case where $\alpha$ and $\beta$ are uniformly distributed over $(0, 1)$ with $\alpha \leq \beta$. The probability that $\beta < \sqrt{\alpha^2 - \alpha + 1}$ is given by: The area of the region $A = (\alpha, \beta) \mid 0 < \alpha \leq \beta \leq 1$ is $\frac{1}{2}$; the area of the region $B = (\alpha, \beta) \in A \mid \beta < \sqrt{\alpha^2 - \alpha + 1}$ is $\int_0^1 \left(\sqrt{\alpha^2 - \alpha + 1} - \alpha\right) d\alpha = \frac{3}{8} \ln 3$. Therefore, the probability is $P = B/A = \frac{3}{4} \ln 3 \approx 0.824$,, indicating that the condition holds in approximately 82.4% of cases. This means that for most parameter choices, it achieves computational efficiency. Moreover, even if the condition is not strictly met in the current block, the reduced sequence length $L'$ propagates to subsequent blocks, ensuring that all following attention computations benefit from the shorter sequence, leading to overall computational savings across the network.

## 3.1 INTEGRATION WITH TRANSFORMER BLOCKS

$$x_l' = \mathrm{MSA}(\mathrm{LN}(x_{l-1})) + x_{l-1}$$

$$x_l'' = \mathrm{MaMe}(x_l')$$

$$x_l = \mathrm{MLP}(\mathrm{LN}(x_l'')) + x_l'' \tag{11}$$

Where $x_{l-1}$ be the token sequence output by block $l-1$, and MSA, MLP, and LN denote Multi-head Self-Attention, Multi-Layer Perceptron, and Layer Normalization, respectively.

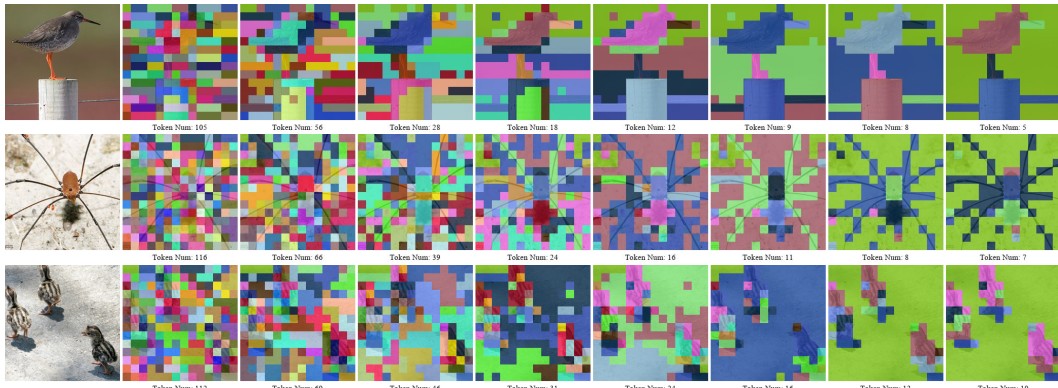

Figure 1: The visualization illustrates the progression of token count reduction in the first 8 blocks of the AugReg ViT-B/16 with MaMe. Each color represents a distinct type of token. See the Appendix 5 for more results.

## 4 EXPERIMENTS

### 4.1 IMAGE CLASSIFICATION

**Training-Free** We evaluate MaMe on DeiT (Touvron et al., 2021) and MAE (He et al., 2022) using pre-trained weights, applying compression to the first 8 layers with a similarity threshold of 0.8. All other things remain identical to (Chen et al., 2023). Table 1 compares token compression methods on ViT models. Table 1 evaluates several token compression methods on ViT models. For ViT-S (DeiT), MaMe achieves 9015 img/s, 79% higher than baseline while maintaining 78.61% accuracy (1.2 points below original), surpassing EViT (8950 img/s, 73.83% accuracy) and ToMe (8874 img/s, 77.99% accuracy). For ViT-B (DeiT), MaMe delivers 4117 img/s (93% faster) with 79.80% accuracy. EViT shows higher throughput (4230 img/s) but lower accuracy (74.61%), while DiffRate(Chen et al., 2023) has similar speed but 78.98% accuracy.

Notably, comparing ViT-B (DeiT) and ViT-B (MAE), which share identical architecture, reveals significant differences in MaMe's performance. On ViT-B (DeiT), MaMe achieves 4117 imgs/s with 79.80% accuracy, while on MAE, throughput increases to 5418 imgs/s, representing a 31.6% throughput improvement, while maintaining 79.83% accuracy. This highlights MaMe's ability to leverage MAE's self-supervised robust representations effectively for token merging. EViT and ToMe show no throughput change between ViT-B models. MaMe's advantage grows with model size: for ViT-L (MAE), it achieves 2764 imgs/s (EViT's 1.63x) while maintaining the highest accuracy (84.81%). On ViT-H (MAE), MaMe delivers 908 imgs/s (almost EViT's 2x), with only a marginal accuracy decrease compared to DiffRate.

**Training-From-Scratch** We integrate MaMe into standard ViT and hierarchical architectures (e.g., Swin(Liu et al., 2021), Iwin (Huo & Li, 2025)) following established training recipes(Liu et al., 2021). For ViT, we apply 2:1 token merging at layers 3, 6, and 9 (similarity threshold: 0.5). For hierarchical models, MaMe replaces downsampling layers, reducing tokens to 25% of the original and eliminating the need for embedding dimension doubling, thus cutting parameters and FLOPs significantly. In order to maintain a regular shape for window partition, we had to discard the "preserved source tokens". Table 3 shows ImageNet-1k results. MaMe boosts throughput across all models with minimal accuracy loss. For example, ViT-T[†] achieves 4462 img/s (vs. 2291 img/s baseline) with only a 1.3% accuracy drop (70.9% vs. 72.2%). ViT-B[†] runs 92% faster (813 img/s) at a 5.8% accuracy cost (76.0% vs. 81.8%).

This exploratory work applies MaMe as downsampling in hierarchical pyramid networks, sacrificing some performance due to architectural constraints that require discarding "preserved source tokens." Despite this, MaMe significantly reduces parameters and computation by eliminating embedding dimension doubling, enabling deployment on edge devices. There is room for improvement, which will be left for future work.

| Model | Method | FLOPs (G) | Throughput (img/s) | Top-1 Acc (%) |
|---|---|---|---|---|
| **Training Free on ImageNet-1K (224×224)** | | | | |
| ViT-S (DeiT) | Baseline | 4.6 | 5039 | 79.82 |
| | EViT | 2.3 | 8950 | 73.83 |
| | ToMe | 2.3 | 8874 | 77.99 |
| | DiffRate | 2.3 | 8875 | 78.75 |
| | MaMe | - | 9015 | 78.61 |
| ViT-B (DeiT) | Baseline | 17.6 | 2130 | 81.83 |
| | EViT | 8.7 | 4230 | 74.61 |
| | ToMe | 8.8 | 4023 | 77.84 |
| | DiffRate | 8.7 | 4124 | 78.98 |
| | MaMe | - | 4117 | 79.80 |
| ViT-B (MAE) | Baseline | 17.6 | 2130 | 83.72 |
| | EViT | 8.7 | 4230 | 75.15 |
| | ToMe | 8.8 | 4023 | 78.86 |
| | DiffRate | 8.7 | 4150 | 79.96 |
| | MaMe | - | 5418 | 79.83 |
| ViT-L (MAE) | Baseline | 61.6 | 758 | 85.95 |
| | EViT | 29.7 | 1672 | 81.52 |
| | ToMe | 31.0 | 1550 | 84.24 |
| | DiffRate | 31.0 | 1580 | 84.65 |
| | MaMe | - | 2764 | 84.81 |
| ViT-H (MAE) | Baseline | 167.4 | 299 | 86.88 |
| | EViT | 99.1 | 512 | 85.54 |
| | ToMe | 92.9 | 500 | 86.01 |
| | DiffRate | 93.2 | 504 | 86.40 |
| | MaMe | - | 908 | 86.16 |

Table 1: Token compression on off-the-shelf models. Throughput measured on an A100 GPU (bs=1024, fp16).

| Model | Method | Input Size (px) | Throughput (img/s) | Top-1 Acc (%) |
|---|---|---|---|---|
| **Zero-Shot Classification on ImageNet-1K** | | | | |
| CLIP (ViT-L/14) | Baseline | 224 | 51.22 | 70.34 |
| | ToMe(r=8) | 224 | 51.33 | 68.98 |
| | ToMe(r=12) | 224 | 52.86 | 66.00 |
| | MaMe($\tau = 0.7$) | 224 | 69.09 | 67.60 |
| | MaMe($\tau = 0.8$) | 224 | 64.01 | 69.95 |
| SigLIP (ViT-B/16) | Baseline | 512 | 46.28 | 75.61 |
| | ToMe(r=32) | 512 | 55.10 | 74.33 |
| | ToMe(r=64) | 512 | 71.94 | 70.66 |
| | MaMe($\tau = 0.8$) | 512 | 79.25 | 71.17 |
| | MaMe($\tau = 0.9$) | 512 | 58.10 | 74.50 |
| SigLIP2 (ViT-B/16) | Baseline | 512 | 43.90 | 78.37 |
| | ToMe(r=32) | 512 | 50.89 | 76.46 |
| | ToMe(r=64) | 512 | 68.07 | 71.60 |
| | MaMe($\tau = 0.9$) | 512 | 76.15 | 75.09 |
| | MaMe($\tau = 0.95$) | 512 | 56.15 | 78.02 |

Table 2: Zero-shot results. Inference throughput measured on a 3090 GPU (bs=1, fp16).

| Model | Param (M) | FLOPs (G) | Throughput (img/s) | Top-1 Acc (%) |
|---|---|---|---|---|
| **Training From Scratch on ImageNet-1K (224×224)** | | | | |
| ViT-T | 5.72 | 1.3 | 2291 | 72.2 |
| ViT-T$^\dagger$ | 5.72 | 0.6 | 4462 | 70.9 |
| Swin-T | 29.0 | 4.5 | 950 | 81.3 |
| Swin-T$^\dagger$ | 1.45 | 1.5 | 1236 | 60.3 |
| Iwin-T | 30.2 | 4.7 | 874 | 82.0 |
| Iwin-T$^\dagger$ | 1.46 | 1.5 | 1522 | 65.1 |
| ViT-S | 22.0 | 4.6 | 1157 | 79.8 |
| ViT-S$^\dagger$ | 22.0 | 2.1 | 2257 | 77.0 |
| Swin-S | 50.0 | 8.7 | 548 | 83.0 |
| Swin-S$^\dagger$ | 2.80 | 1.8 | 1043 | 65.8 |
| Iwin-S | 51.6 | 9.0 | 512 | 83.4 |
| Iwin-S$^\dagger$ | 2.82 | 1.8 | 1254 | 71.0 |
| ViT-B | 86.4 | 17.6 | 422 | 81.8 |
| ViT-B$^\dagger$ | 86.4 | 8.4 | 813 | 76.0 |

Table 3: Comparative evaluation with MaMe compression ($^\dagger$). Throughput measured on an A100 GPU (bs=64, fp32).

| Backbone | UperNet 160k | | | |
|---|---|---|---|---|
| | Param(M) | FLOPs(G) | mIoU(%) | mAcc(%) |
| Swin-T | 59.9 | 945 | 44.5 | 55.6 |
| Swin-T$^\dagger$ | 28.4 | 432 | 33.1 | 44.1 |
| Iwin-T | 61.9 | 946 | 44.7 | 56.6 |
| Iwin-T$^\dagger$ | 28.5 | 432 | 26.1 | 35.0 |
| Swin-S | 81.3 | 1038 | 47.6 | 58.8 |
| Swin-S$^\dagger$ | 29.8 | 435 | 35.8 | 47.0 |
| Iwin-S | 83.2 | 1038 | 47.5 | 59.3 |
| Iwin-S$^\dagger$ | 29.8 | 435 | 31.6 | 41.6 |

Table 4: Results for ADE20K semantic segmentation. FLOPs measured with input size 512×2048.

| Model | Method | Input (FxHW) | Throughput (videos/s) | Top-1 Acc (%) |
|---|---|---|---|---|
| **Action Recognition on Kinetics-400** | | | | |
| VideoMAE-B | Baseline | 16x224 | 13.24 | 76.81 |
| | ToMe(r=96) | 16x224 | 13.76 | 75.54 |
| | ToMe(r=128) | 16x224 | 14.06 | 73.34 |
| | MaMe($\tau = 0.85$) | 16x224 | 13.81 | 74.23 |
| | MaMe($\tau = 0.9$) | 16x224 | 13.33 | 76.03 |
| VideoMAE-L | Baseline | 16x224 | 6.25 | 82.31 |
| | ToMe(r=32) | 16x224 | 6.97 | 82.05 |
| | MaMe($\tau = 0.8$) | 16x224 | 9.28 | 81.47 |

Table 5: Results of VideoMAE on action recognition benchmarks. Inference throughput is measured on a 3090 GPU (bs=1, fp16).

## 4.2 SEMANTIC SEGMENTATION

To assess our compressed models to downstream dense prediction tasks, we evaluate the Swin and Iwin backbones on the ADE20K (Zhou et al., 2019) using UperNet (Xiao et al., 2018) in MM-Segmentation (Contributors, 2020). Following (Liu et al., 2021) settings, results in Table 4 show MaMe marked as ($^\dagger$) reduces model complexity, with Swin-T$^\dagger$ decreasing parameters by 52% and FLOPs by 54% versus baseline, though with reduced segmentation performance. While compressed

Iwin$^\dagger$ models achieve strong classification accuracy (Iwin-S$^\dagger$ at 71.0%), their semantic segmentation performance drops more than compressed Swin models, with Iwin-S$^\dagger$ declining 15.9 points versus Swin-S$^\dagger$'s 11.8 points.

## 4.3 Multimodal Large Language Models

**Zero-shot Image Classification** We conducted zero-shot image classification on ImageNet-1K validation set to evaluate token merging strategies across CLIP(Radford et al., 2021), SigLIP(Zhai et al., 2023), and SigLIP2(Tschannen et al., 2025). For CLIP, MaMe ($\tau = 0.8$) increased throughput by 25% (64.01 img/s) with 0.39% accuracy drop, while ToMe (r=12) gave 3% throughput gain with 4.34% accuracy loss. For SigLIP, MaMe ($\tau = 0.9$) improved throughput by 25% (58.10 img/s) with 1.11% accuracy reduction, while ToMe (r=32) achieved 19% speedup with 1.28% accuracy loss. For SigLIP2, MaMe ($\tau = 0.95$) increased throughput by 28% (56.15 img/s) with 0.35% accuracy drop, while ToMe (r=32) gave 16% throughput gain with 1.91% accuracy loss. SigLIP2's ability to merge tokens at $\tau = 0.95$ indicates its confident semantic representations. MaMe demonstrates better balance between throughput and accuracy versus baseline and ToMe.

**Text-Image to Text** We evaluated the impact of token merging on the LLaVA-1.5-7B model (Liu et al., 2023a) using the VLMEvalKit framework (Duan et al., 2024) across various multimodal benchmarks (Fu et al., 2023; Yue et al., 2023; Lu et al., 2022; Li et al., 2023; Lin et al., 2024; Liu et al., 2024; 2023b), Merging was applied to the visual encoder to reduce the number of visual tokens fed to the language model. We compare the baseline against ToMe with a fixed reduction ratio of r=8 per layer and MaMe with a similarity threshold of $\tau = 0.8$. As shown in Table 6, both methods significantly reduce evaluation time. MaMe achieves greater acceleration while delivering metric scores that are competitive with or slightly better than ToMe on most of benchmarks. This demonstrates that token merging, particularly MaMe, is a highly effective strategy for accelerating large multimodal models with minimal impact on performance.

| Method | MME | | MMMU | | ScienceQA | | SEED-Image | | MMStar | | CRPE | | MMBench | |
|---|---|---|---|---|---|---|---|---|---|---|---|---|---|---|
| | Metric | Time(s) | Metric | Time(s) | Metric | Time(s) | Metric | Time(s) | Metric | Time(s) | Metric | Time(s) | Metric | Time(s) |
| LLaVA-1.5-7B (Baseline) | 32.76 | 597 | 32.22 | 481 | 65.43 | 625 | 60.17 | 3513 | 32.53 | 565 | 50.69 | 2076 | 62.80 | 1191 |
| + ToMe (r=8) | 31.40 | 509 | 30.11 | 440 | 63.42 | 554 | 58.19 | 3135 | 31.13 | 545 | 46.78 | 1794 | 61.00 | 1086 |
| + MaMe ($\tau$=0.8) | 31.40 | 447 | 30.56 | 422 | 64.47 | 478 | 57.20 | 2840 | 30.27 | 531 | 45.62 | 1659 | 60.48 | 1020 |

Table 6: Benchmark results for LLaVA-1.5-7B with different token merging methods. For each benchmark, we report the primary **Metric** (e.g., accuracy) and the total evaluation **Time** in seconds.

## 4.4 Video Classification

We apply token merging to VideoMAE (Tong et al., 2022) models' vision encoder and compare MaMe with ToMe on Kinetics-400 validation set (Kay et al., 2017). We sample 16 frames at $224 \times 224$ resolution per video clip. We report Top-1 accuracy and inference throughput in videos/s on a 3090 GPU with fp16 precision. Results in Table 5 show token merging effectively accelerates video transformer inference. For VideoMAE-B, MaMe with threshold $\tau = 0.9$ increases throughput from 13.24 to 13.33 videos/s with only 0.78% drop in Top-1 accuracy, outperforming ToMe(r=128) which shows 3.47% accuracy degradation for similar speedup.

## 4.5 Ablation Study

### 4.5.1 Algorithmic Design Choices

Our ablation studies the core algorithmic components of MaMe, with results summarized in Table 2.

**Feature Choice:** The raw token matrix `x` achieves optimal accuracy (83.35%) with competitive throughput. Features `k` and `k-mean` show lower accuracy (71.63% and 69.01%), confirming `x` preserves the most discriminative information for merging decisions.

**Similarity Function:** Cosine similarity achieves the best balance (83.35% accuracy, 73.06 im/s throughput). Dot product improves throughput (81.14 im/s) but sacrifices accuracy (80.43%), while Euclidean and softmax-based methods underperform in either metric.

**Partition Style:** Sequential ordering maximizes accuracy (84.14%) but reduces throughput (71.81 im/s). Alternating order offers the best trade-off (83.35% accuracy, 73.06 im/s), outperforming random ordering, which slightly boosts throughput at the cost of accuracy.

**Adaptive Weight Pruning:** AugReg models require pruning to achieve best 83.35% accuracy. MAE models show moderate gains but remain less dependent on pruning.

| feature | acc | im/s | | function | acc | im/s | | order | acc | im/s | | src | pruning | acc | im/s |
|---|---|---|---|---|---|---|---|---|---|---|---|---|---|---|---|
| x | **83.35** | 73.06 | | eucl | 81.58 | 73.60 | | sequential | 84.14 | 71.81 | | mae | | 77.51 | 85.59 |
| k | 71.63 | 72.45 | | cosine | 83.35 | 73.06 | | alternating | 83.35 | 73.06 | | mae | ✓ | 80.02 | 78.96 |
| k-mean | 69.01 | 75.06 | | dot | 80.43 | 81.14 | | random | | 83.24 | 73.74 | | augreg | | 72.47 | 78.59 |
| | | | | softmax | 61.90 | 80.33 | | | | | | augreg | ✓ | 83.35 | 73.06 |

(a) **Feature Choice.** The x matrix has the most information within tokens.

(b) **Similarity Function.** Cosine similarity is the best choice for speed and accuracy.

(c) **Partition Style.** Alternating is more reliable and faster.

(d) **Adaptive Weight Pruning.** AugReg models need pruning.

Figure 2: Ablation experiments using AugReg ViT-B/16. Our default settings are marked in Gray. We report Top-1 accuracy (acc) with FP32 precision and model inference throughput (im/s) on a 3090 GPU. The visualization results of different methods are shown in Figure 3.

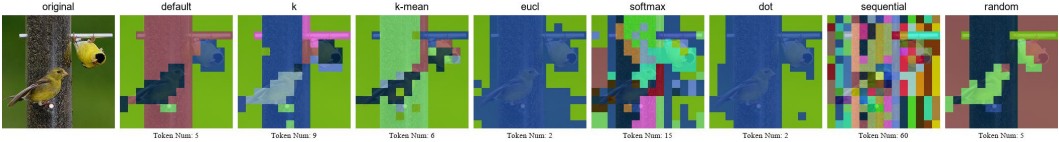

Figure 3: The visualization in the 8th block of the AugReg ViT-B/16 using MaMe with different settings. Each color square represents a distinct type of token. Default is our default method.

### 4.5.2 WHERE AND WHAT

To investigate where MaMe should be applied within models and what similarity threshold yields optimal performance, we examine the joint impact of similarity threshold ($\tau$) and the number of blocks applying token merging ($num\_block$) on both accuracy and throughput on ViT models as shown in Figure 4.

**Accuracy** The relationship between similarity threshold and accuracy shows a non-linear, saturating pattern across ViT architectures, influenced by MaMe applied block depth. As the threshold increases from 0.6 to 0.8, accuracy improves rapidly before plateauing, indicating diminishing returns from stricter token retention and suggesting that exceeding a critical threshold sufficiently distinguishes features.

**Throughput** Throughput monotonically decreases with similarity threshold, dropping sharply at low thresholds due to rising computational costs. Threshold sensitivity inversely correlates with model size: ViT-S experiences the steepest decline, followed by ViT-B and ViT-L, indicating larger models better mitigate merging overhead.

**Model-Scale Sensitivity** Model sensitivity to token merging varies by size: ViT-L maintains >85% accuracy across thresholds (0.6–0.8) with throughput gains, ViT-B shows moderate sensitivity, and ViT-S is most vulnerable (accuracy drops from ~79% to ~60% with aggressive merging). Larger models exhibit greater representational redundancy, allowing coarser merging with minimal performance loss.

**Random Partition** The comparison between deterministic default configurations (dashed lines) and stochastic trials (scatter points) indicates that the default settings define a Pareto frontier: stochastic partitions yield higher accuracy (points above the dashed line) but lower throughput (points below the dashed line). This trade-off suggests that while stochasticity enhances accuracy, it undermines computational efficiency. The deterministic, alternating partition thus serves as a robust baseline, balancing performance and efficiency.

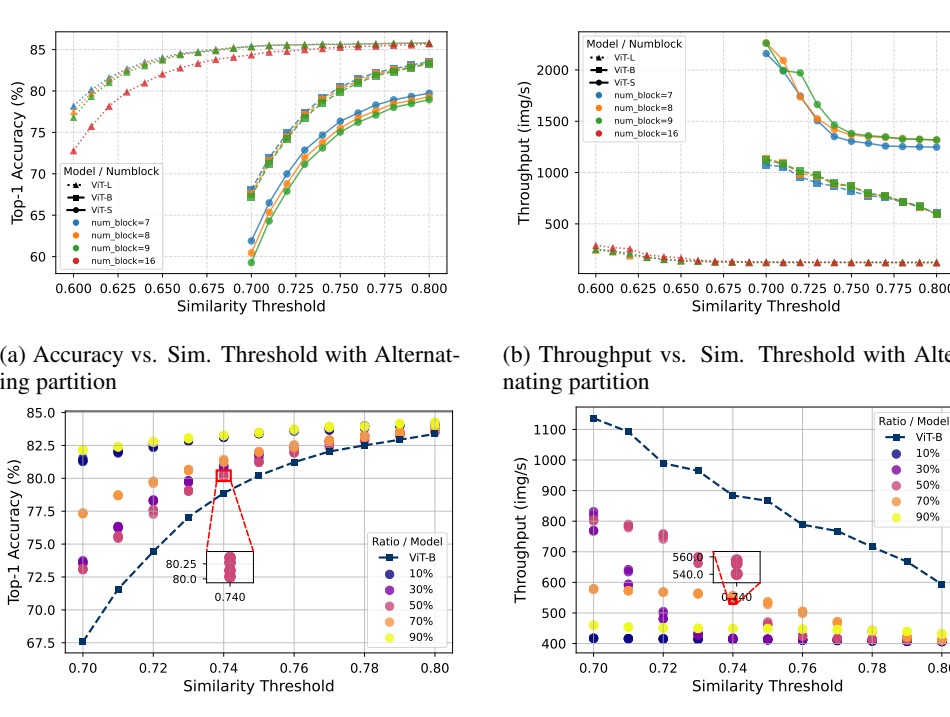

(a) Accuracy vs. Sim. Threshold with Alternating partition

(b) Throughput vs. Sim. Threshold with Alternating partition

(c) Accuracy vs. Sim. Threshold with Random partition

(d) Throughput vs. Sim. Threshold with Random partition

Figure 4: The accuracy and throughput change with the similarity threshold under both alternating and random partitions. Five different random seeds are employed to conduct five experiments under different ratios of tokens as source tokens as shown in the box in the Figures (c) and (d), illustrating that the default, determined alternating partition curves represent a Pareto frontier.

## 5 DISCUSSION

**Pros and Cons** One of the primary advantages of MaMe is its non-intrusive nature with respect to standard attention mechanisms. Unlike ToMe, which requires attention modifications, MaMe preserves the standard attention calculation and easily integrates with optimized implementations like Flash Attention(Dao, 2024). Its full-matrix operations are GPU-friendly, avoiding hard-to-optimize sorting operations. However, the optimal similarity threshold ($\tau$) and layer selection require manual determination. Future work will automate them through adaptive strategies or reinforcement learning.

**Migrating to LLMs** Token merging approaches like ToMe break causality in autoregressive LLMs by allowing tokens to merge with future tokens. MaMe enables **causal token merging** by partitioning tokens into odd (destination) and even (source) sets, then applying a causality mask $M$ ($M_{i,j} = 1$ if $j \leq i$, else 0) to zero upper triangular part of fusion weights $W_{ij}^{\mathrm{F}}$ by $W_{ij}^{\mathrm{F}} \odot M_{ij}$. This restricts each destination token (e.g., token 5) to merge only with preceding source tokens (e.g., tokens 0, 2, 4), preserving causality and allowing reduce KV cache length in LLMs.

## 6 CONCLUSION

In this work, we introduced MaMe, a differentiable, training-free token merging method based on full-matrix operations. Through extensive experimentation, including accelerating off-the-shelf and from-scratch trained ViT models, as well as video classification models, we demonstrated that MaMe, like ToMe, achieves significant inference speedups, typically with a trade-off in performance metrics. Lastly, we discussed MaMe's potential for application in large language models to compress KV cache while preserving causality.

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

# A APPENDIX

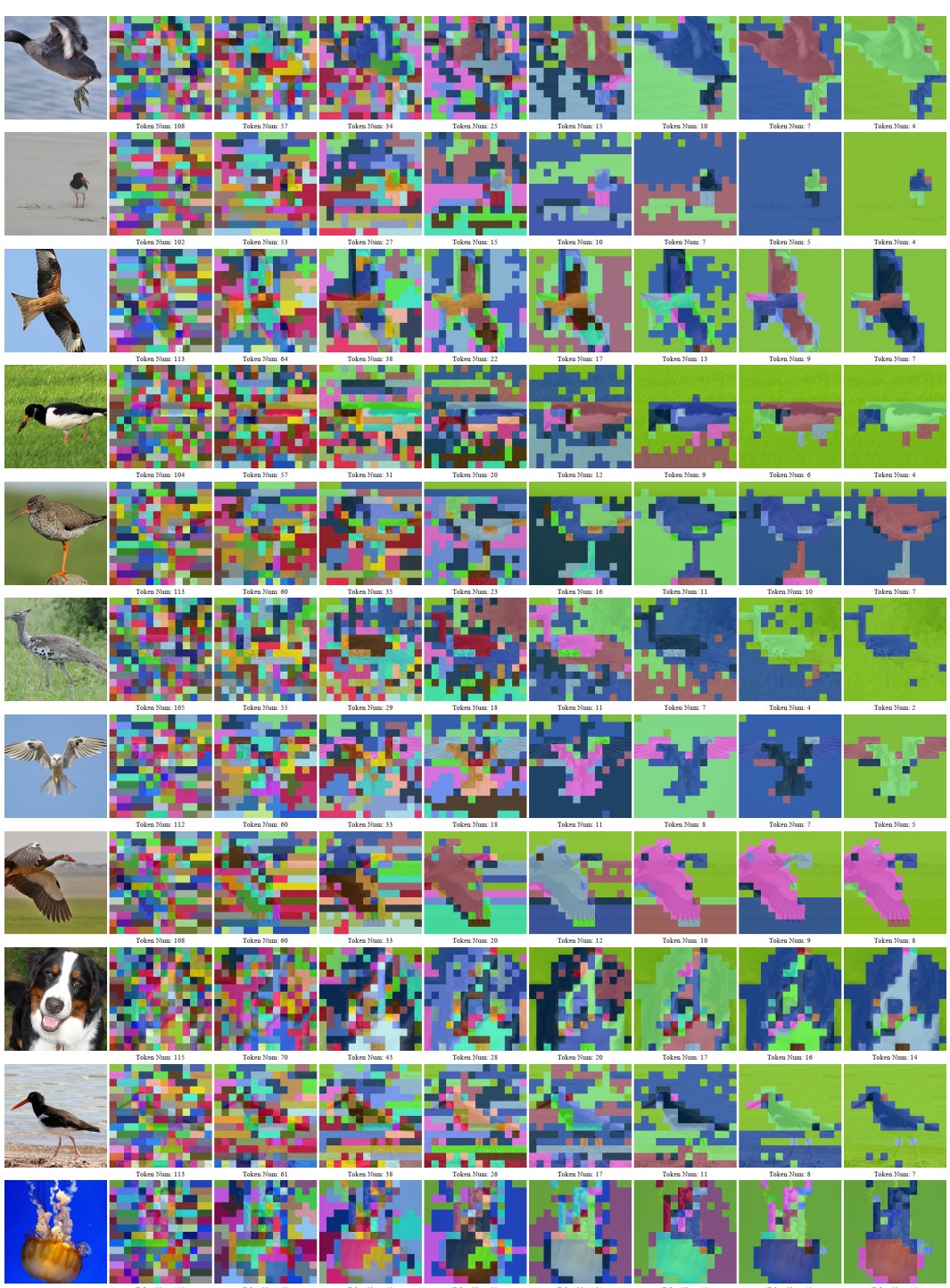

Figure 5: The visualization illustrates the progression of token count reduction in the first 8 blocks of AugReg ViT-B/16 with MaMe. Each color represents a distinct type of token.

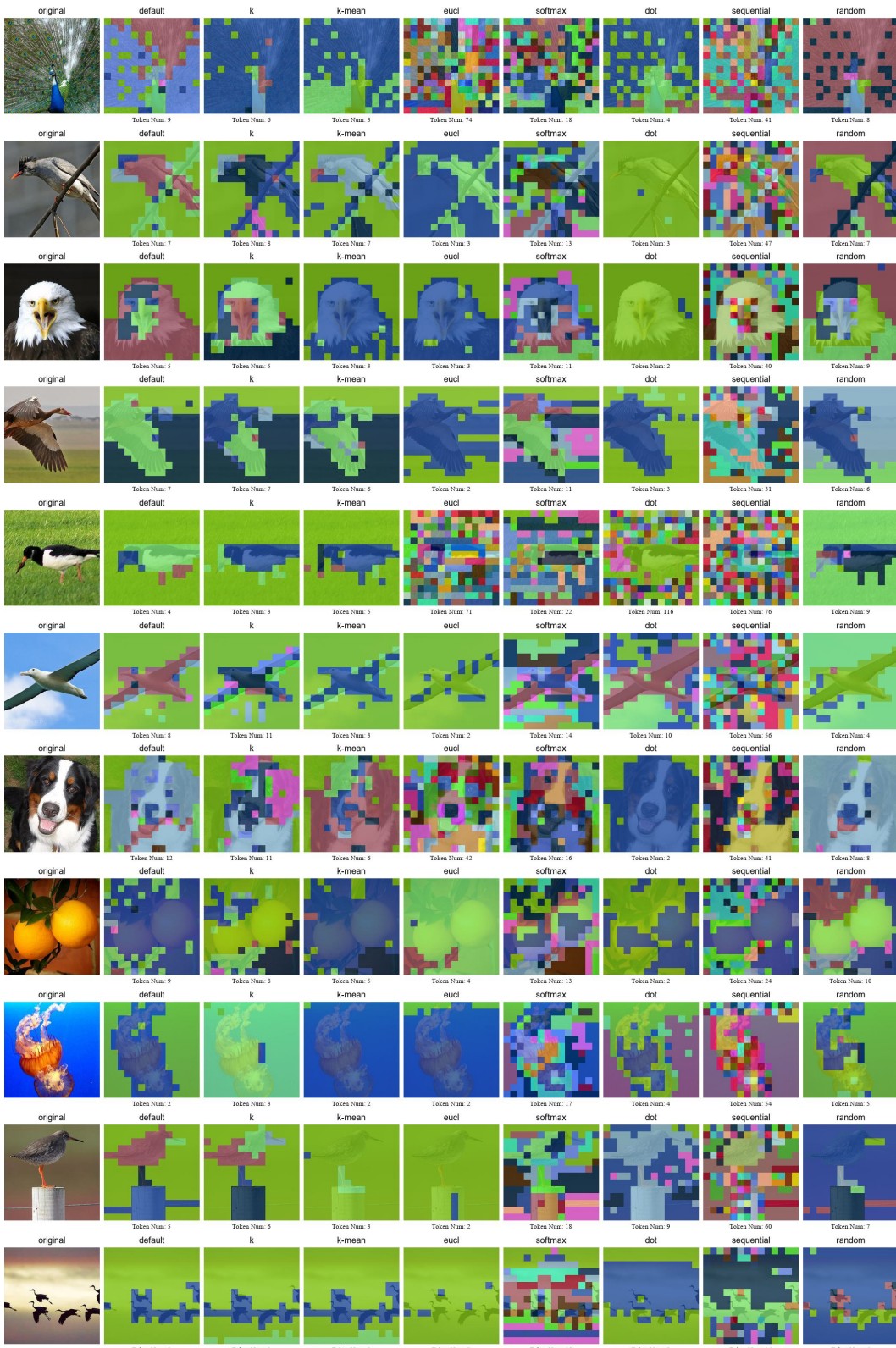

Figure 6: The visualization in the 8th block of the AugReg ViT-B/16 using MaMe with different settings. Each color square represents a distinct type of token. Default is our default method.

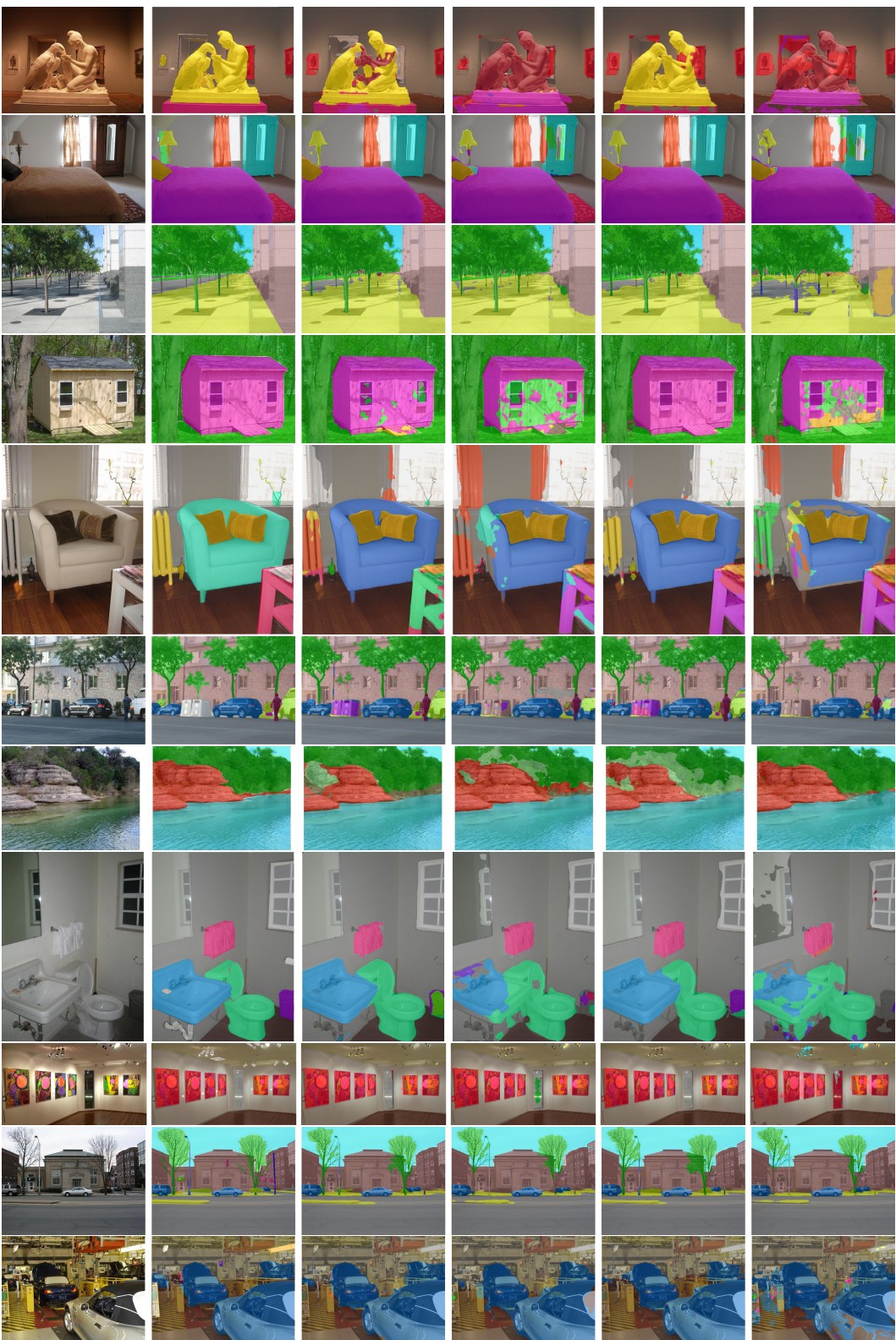

Figure 7: Visualization of semantical segmentation results. From left to right, input image, Ground Truth, Swin, Swin-T[†], Iwin, and Iwin-T[†].

LARGE LANGUAGE MODEL USAGE DECLARATION

In the preparation of this work, the authors utilized several large language models (LLMs) for specific tasks as detailed below. The authors are solely responsible for the content of the publication.

- **Literature Review Assistance:** Gemini 2.5 Pro was employed to assist in gathering and synthesizing relevant research literature. This assistance was primarily used in the preparation of the **Introduction** and **Related Work** sections to identify key developments and contextualize our contribution within the existing body of research.
- **Language Polishing:** Gemini 2.5 Pro, Gemini 2.5 Flash, and DeepSeek-R1 were used to refine English expression throughout the manuscript. This included improving grammatical accuracy, enhancing clarity of technical descriptions, and ensuring consistent academic tone.
- **Experimental Analysis Support:** Gemini 2.5 Pro was utilized as an analytical tool to assist in the interpretation of selected experimental results, particularly in identifying patterns and generating preliminary insights that were subsequently rigorously verified and expanded upon by the authors.
- **Declaration:** Deepseek-R1 was used to assist in writing the declaration.

All content generated with LLM assistance was carefully reviewed, critically evaluated, and substantially modified by the authors to ensure accuracy, originality, and adherence to scientific standards. The final manuscript represents the authors' own intellectual contribution and perspective.

