# OpenReview forum: "MaMe: Matrix-Based Token Merging"
_ICLR.cc/2026/Conference — ICLR 2026 Conference Withdrawn Submission_

### Official Review · Reviewer_rKrx · 2025-10-20

**Soundness:** 3
**Presentation:** 3
**Contribution:** 3
**Rating:** 6
**Confidence:** 4

**Summary:**

The paper addresses the computational inefficiency of Vision Transformers by proposing MaMe, a training-free token merging method based entirely on matrix operations. The core method involves partitioning tokens into source and destination sets, computing cosine similarity matrices, applying adaptive weight pruning, and aggregating tokens while preserving unique ones. Experimental results across image classification, semantic segmentation, multimodal reasoning, and video tasks demonstrate MaMe's effectiveness in accelerating diverse transformer-based models with minimal performance degradation.

**Strengths:**

1. MaMe eliminates non-differentiable operations through matrix-based similarity computation and adaptive pruning, enabling end-to-end training compatibility.

2. MaMe introduces no learnable parameters, maintaining original model complexity and allowing direct integration into pre-trained models without retraining.

3. The method relies on matrix operations with O(MN) complexity per layer, avoiding costly clustering algorithms like k-means, and scales to large models and high-resolution inputs.

4. I really appreciate that the evaluations are quite comprehensive across different domains.

**Weaknesses:**

1. Despite several improvement, the paper seems to be incremental compared to ToMe, which greatly reduced the novelty of the proposed MaMe. Also, the baselines are quite few and out-of-date, and more recent baselines are supposed to be incorporated.

2. Despite the promising O(MN) complexity, mainstream method like ToMe, ATS and so on also does not rely on cluster, so this contribution would be somewhat overclaimed form my humble perspective.

3. The results are not impressive. In table 3 and table 4, the improvement is marginal and in table 6, the reduction in  evaluation time seems at the cost of declining accuracy. More experiments are supposed to be provided to further enhance the method.

4. The computational cost of matrix operations (e.g., similarity calculation) is not quantified relative to baseline attention, potentially underestimating overhead for large token counts.

5. The greedy token partitioning strategy and the fusion matrix lack theoretical guarantees on optimality compared to combinatorial search, and the impact of partition choices is empirically evaluated but not analytically derived.

6. While alternating partition is default, the paper does not ablate other partitioning schemes.

**Questions:**

1.  Would theoretical analysis strengthen the justification for the alternating partition approach and the fusion matrix?

2. Why FLOPs for MaMe is not reported in table 1?

---

> ### Author Response · Authors · 2025-11-14
> **MaMe's advantages**
>
> Thank you for the review and kind words. I truly appreciate you recognizing my work.
>
> 1. Non-Invasive Integration: MaMe integrates seamlessly with standard attention mechanisms, which means MaMe can enjoy the benefit of efficient attention like Flash Attention, ensuring broad and flexible compatibility across models. But ToMe requires modifications to core attention computation; it isn't so plug-and-play.
>
> 2. Preservation of Spatial Locality: MaMe inherently maintains the spatial positional relationships among tokens. This property allows MaMe to be a downsampling layer, eliminating the dimension doubling when spatial reduction. While current empirical performance might show a slight decrease, the significant reduction in parameters and computational load achieved by MaMe makes it exceptionally beneficial for deployment on resource-constrained edge devices. Future refinements are expected to further enhance its performance. ToMe can not do it https://github.com/facebookresearch/ToMe/issues/15.
>
> 3. Causal Merging for LLMs: MaMe supports causal token merging by partitioning tokens into odd (destination) and even (source) sets, then applying a causality mask $M$ ($M_{i,j} = 1$ if $j\leq i$, else $0$) to zero the upper triangular part of fusion weights $W_{ij}^{\text{F}}$ by $W_{ij}^{\text{F}} \odot M_{ij}$. This restricts each destination token (e.g., token 5) to merge only with preceding source tokens (e.g., tokens 0, 2, 4), offering potential to reduce KV cache length in LLMs, optimizing memory and computational efficiency for sequential tasks. ToMe can not do it https://github.com/facebookresearch/ToMe/issues/38.
>
> 4. GPU-Friendly Operations: Matrix makes MaMe GPU-friendly, enhancing inference speed and resource utilization.
>
> MaMe's FLOPs in Table 1 are not reported due to the variable number of preserved tokens; an approximate estimate can be obtained by assuming no tokens are preserved after merging. MaMe is a dynamic token merging method. Batch processing is enabled by using batch processing technology at reviewer qhk5. Furthermore, our experiments also explored alternative partitioning methods in the original paper, and their detailed analyses are discussed in the revised version of the paper.

---

### Official Review · Reviewer_7A9g · 2025-10-26

**Soundness:** 2
**Presentation:** 2
**Contribution:** 2
**Rating:** 2
**Confidence:** 5

**Summary:**

The authors propose Matrix-Based Token Merging, a token merging mechanism for vision transformers and related approaches, aiming to increase throughput while being quality neutral. The authors propose bifurcating tokens into source and destination ones and construct a non parametric weight matrix that updates the destination tokens based on the cosine similarity between the two sets. They demonstrate efficiency driven results on tasks like image classification, action recognition, zero-shot metrics on standard datasets and benchmark against some other related efficient methods like ToMe.

**Strengths:**

- The authors propose a unique approach to token merging -- bringing attention-like ideas between self-defined source and destination tokens to update destination tokens and supress source tokens.
- The authors present a comprehensive literature survey, clearly illustrating the previous relevant works and their shortcomings
- The authors have a well presented section on methodology (barring some concerns listed in the weaknesses section of my review)
- The authors prepare and present a comprehensive set of results on various benchmarks, clearly depicting relevant metrics like throughput, accuracy, etc.

**Weaknesses:**

Please note that this section includes questions too.

- The results indicate sharp drops in accuracy across various benchmarks, even though it comes at a throughput increase.
  - For instance, in Table 1, ViT-B (Deit) with MaMe improves over the baseline in terms of throughput, but at an accuracy cost, which when compared with the baseline for ViT-S (Deit), the throughput is lower at a neutral accuracy.
  - Similarly, Table 3 indicates that even when training with the MaMe setup from scratch, the performance remains quite low as compared to the baseline variants for ViT and SWIN.
  - Similar observation for segmentation in Table 4
  - Why is the throughput increase so small for action recognition (K6400) in Table 5.

- The methodology has some unclear things, and is not backed up by ablations well.
  - Why is random, alternating or any other bifurcation intuitively logical for separating source and destination tokens, other than empirical evidence.
  - It is not explained why there are two rounds of filtering and normalization, once at equation 3, and then again at equation 7.
  - Why is equation 6 augmented with an $\epsilon$ which might lead to an inaccurate computation of the count, why not do $\sum_i W_{ij} > 0$.
  - $X_{pres}$ in equation 10 is not defined earlier. It is not clear why or how this is constructed. Similarly, the "merge" operation in equation 11 is not well defined.
  - Several of these design choices are not intuitively obvious and need ablations to support why these steps are necessary or contribute to the method's effectiveness.

On a overall level, the paper will benefit from a more thorough treatment of the methodology, justifying the design choices, and working on stronger accuracy results while attempting to improve throughput.

**Questions:**

Please see the section above.

---

> ### Author Response · Authors · 2025-11-14
> **Explanation**
>
> Thank you for the review. I understand your concern about how these formulas were ultimately conceived. These are the results of my continuous revisions and improvements over a long period of time, a process that cannot be written down in a paper.
>
> For differentiability, we don't apply boolean operations $C_{j}= \sum_{i=1}^{M} W_{ij}>0$.
>
> Some throughput results are obtained by calculating the average verification time when the batch size is 1.
>
> The reasons for training MAME from scratch on Swin have been explained elsewhere. Training MAME from scratch on VITs has demonstrated that it significantly improves training speed and throughput with minimal performance degradation. It's possible to achieve performance exceeding baseline through fine-tuning, similar to that described in the ToMe paper. Training from scratch and fine-tuning are not the same. Please refer to the revised PDF for other questions.

---

### Official Review · Reviewer_qhk5 · 2025-10-30

**Soundness:** 2
**Presentation:** 1
**Contribution:** 2
**Rating:** 2
**Confidence:** 4

**Summary:**

This paper presents MaMe, a method to merge tokens in Vision Transformers (ViTs) to improve efficiency. The paper claims that this method brings four key innovations: (1) it is fully differentiable to allow ViTs to be trained better from scratch, (2) it uses efficient matrix operations unlike existing approaches, to achieve a larger speedup, (3) it is parameter-free, simplifying deployment and reducing complexity, and (4) it can be applied directly to pre-trained ViTs. MaMe works by dividing tokens into two sets A and B, calculating the cosine similarity with all tokens in set B for each token in set A, and merging tokens from set B using a weighted average if their cosine similarity is above a certain threshold. With experiments on image classification, applying MaMe directly to pre-trained ViTs, MaMe is shown to achieve a better speed-accuracy balance than existing token reduction approaches. When training a ViT with MaMe from scratch, the model performs worse than when applying MaMe to pretrained models. MaMe achieves mixed results on other tasks. It achieves a comparable speed-accuracy trade-off to existing token reduction methods for action recognition and several vision-language tasks, but causes large accuracy drops when applied to semantic segmentation models.

**Strengths:**

1. By focusing on token reduction in ViTs, the paper tackles a relevant problem. If an effective strategy is found to reduce the number of tokens processed by a ViT without causing a drop in accuracy, then the efficiency of the ViT would improve without compromising task performance. This could make the ViT model more suitable for various applications.

2. The proposed method, MaMe, obtains a better speed-accuracy balance than existing token merging approaches in several settings. As shown in Tab. 1, MaMe obtains a higher accuracy and throughput than existing approaches EViT, ToMe and DiffRate across various model sizes and training strategies (DeiT & MAE). Similar improvements are obtained for zero-shot image classification in Tab. 2.

3. The visualization of the progression of token reduction in Fig. 2 provides interesting insights in the effect of the token merging operations across different ViT layers. The visualization shows that MaMe is effective in merging tokens that belong to similar semantic categories, which is the intended behavior.

**Weaknesses:**

1. There is no text in Sec. 4.5 (titled Ablation Study). There are several figures and tables with additional results (Fig. 3, Fig. 4, Tab. 7), but these are not referred to in the paper. As a result, it is not clear what can be seen in these figures and tables, what the significance is of these results, and what conclusions can be drawn from them. Overall, the 'empty' subsection harms the presentation and readability of the paper.

2. The technical innovation of the proposed MaMe method is limited. In its core, MaMe is a slightly more elegant, matrix-based implementation of methods that merge tokens when the cosine similarity of token embeddings is above a certain threshold, such as ALGM [a] for semantic segmentation. Like MaMe, the global merging module of ALGM also splits up tokens into two sets (like ToMe [b]), then compares each token in set A with each token in set B, and merges tokens if their cosine similarity is above a certain threshold. Like MaMe, ALGM and ToMe are also parameter-free and 'plug-and-play', so these claimed key innovations (L055-L069) are not unique. Comparing ALGM and MaMe, the only key operational difference is in the way the merged token embeddings are obtained, taking a simple average in ALGM and using a weighted average and summing with 'destination tokens' in MaMe. To properly demonstrate the value of MaMe, and show that the small nuanced differences between MaMe and existing methods constitute meaningful and effective innovations, the paper could (a) explain in text how MaMe differs conceptually from existing methods like ALGM and ToMe, and (b) compare quantitatively to existing threshold-based merging methods like ALGM.

3. The paper does not demonstrate the positive impact of the fact that MaMe is fully differentiable, while this is a key contribution/innovation of the paper. The claim (L057-L060) is that MaMe being fully differentiable makes training from scratch more optimal. However, comparing Tab. 3 to Tab. 1, when training a ViT from scratch with MaMe, the accuracy drop from the baseline is significantly larger than when applying MaMe directly to a pre-trained ViT. In contrast, 'non-differentiable' method ToMe [b] *performs better* when trained from scratch, as shown in their original paper. These findings all contradict the claim/hypothesis that MaMe being fully differentiable is beneficial for training from scratch. Since this is one of the key contributions of the paper, this is a considerable weakness. To improve this, the claim should either be amended or supported with other experimental results.

4. The abstract (L015) claims that Swin-S with MaMe obtains 47.0 mIoU on ADE20K, but this number does not correspond with the results from Tab. 4 (35.8 mIoU and 47.0 mAcc). Thus, this claim is unjustified and should be amended.

5. Applying MaMe to Swin or Iwin leads to very large drops in accuracy. This is the case for both image classification (Tab. 3) and semantic segmentation (Tab. 4). These large accuracy drops make that MaMe will not be useful in those settings, in practice. These results show that the presented approach is not generally applicable to various Transformer-based backbones, which reduces the value of the work.

6. The paper does not explain why the number of parameters decreases when applying MaMe to Swin and Iwin, in Tab. 3 and Tab. 4. If I understand the paper correctly, MaMe only reduces the number of tokens that are processed by the model. How is it possible that the number of model parameters also decreases as a result? This should be explained.

7. The paper does not describe how it handles batching, in case different numbers of tokens are merged for different images in a batch. For instance, in Tab. 1, MaMe uses a threshold of 0.8 (L244), which can result in a different number of tokens being merged for each image, and the experiment uses a batch size of 1024 (L350). How are images batched in case the resulting number of tokens per image is different, due to threshold-based merging? Is any padding being used, which would negate some of the effect of token reduction? This should be explained.

8. Several implementation details are not provided, which harms reproducibility. For instance, the paper does not mention (a) how the *source* and *destination* token sets are obtained (L148 only mentions that the strategy can very), and (b) what the hyperparameters are (learning rate, batch size, optimizer, etc.) for training the models in Tab. 3 and Tab. 4 from scratch. It would help if the paper had a separate subsection/paragraph in which these implementation details are specified.

9. Tab. 1 does not provide the FLOPs results for the proposed MaMe method, while it does provide these results for the baseline and other token reduction methods. This means that there is not a full picture of the efficiency of MaMe compared to existing work, making it difficult to fully judge its performance.

[a] Norouzi et al., ALGM: Adaptive Local-then-Global Token Merging for Efficient Semantic Segmentation with Plain Vision Transformers, CVPR 2024.
[b] Bolya et al., Token Merging: Your ViT But Faster, CVPR 2023.

**Questions:**

As the paper has a considerable number of weaknesses (see above) and relatively minor strengths, I give a *reject* rating. Some of the issues could be fixed relatively easily (e.g., by having a proper ablation study and providing implementation details), but most other issues are more fundamental (e.g., the limited technical innovation of the method and the poor performance in some settings). In the 'weaknesses' section above, I have provided several suggestions to resolve these issues. Beyond the suggestions and questions posed there already, I have no additional questions at this point.

---

> ### Author Response · Authors · 2025-11-14
> **Batch Processing Implementation.**
>
> Thank you for your very thorough and responsible review comments. Due to page limitations, many points were not described in detail. Now you can refer to the revised version.
>
> For efficient implementation on batched data, the preservation decision must be consistent across all samples in a batch. Given the fusion matrix for a batch be $\mathbf{W}^F \in \mathbb{R}^{B \times M \times N}$. A per-sample preservation mask $m^{(b)} \in \{0,1\}^N$ is computed for each sample $b$, where $m^{(b)}j = \mathbb{I}(\sum_{i=1}^{M} W^F_{b,i,j} = 0) $. To ensure batch consistency, a source token $j$ is preserved if it is marked for preservation in \textit{any} sample, yielding a final batch-wide mask $m^{\text{final}}j = \bigvee{b=1}^{B} m^{(b)}j$. So the preserved source tokens are $\mathbf{X}{\text{pres}}=\{\mathbf{x}{j}^{src} \mid m^{\text{final}}_j=1\}$. Subsequently, to prevent preserved tokens from fusing, the corrected fusion matrix $\mathbf{\tilde{W}}^{F}$ is obtained by $\mathbf{\tilde{W}}^{F}{b, i,j} = \mathbf{W}^F{b, i,j} \cdot (1 - m^{\text{final}}_j)$, zeroing out columns corresponding to preserved tokens and keeping others unchanged.

---

> ### Author Response · Authors · 2025-11-14
> **MaMe as downsampling layer**
>
> For hierarchical pyramid structures like Swin Transformer, we replace the original downsampling layers with MaMe by setting 25\% tokens as destination tokens, reducing the token count to $\frac{1}{4}$ of the original. In order to maintain a regular shape for the window partition, we had to discard the "preserved source tokens". ToMe is unsuitable for this task as its reliance on sorted similarity rankings would disrupt spatial relationships. This modification eliminates the need for embedding dimension doubling after downsampling, maintaining consistent feature dimensions throughout the network and reducing parameters and flops dramatically, facilitating deployment on edge devices.
>
> This is a feasibility exploratory experiment because the discarded preserved tokens and the fact that the embedding dimension does not increase after space reduction may limit the model's performance. However, there is room for improvement, such as setting a learnable similarity threshold or further increasing the embedding dimension after merging using MLP, etc.

---

### Official Review · Reviewer_1TaD · 2025-10-31

**Soundness:** 3
**Presentation:** 3
**Contribution:** 3
**Rating:** 6
**Confidence:** 3

**Summary:**

The paper presents a novel, training-free method for reducing the computational cost of Vision Transformers (ViTs) by merging similar tokens. The core innovation is a differentiable, parameter-free algorithm that uses efficient matrix operations for token fusion, addressing important limitations of existing methods like non-differentiability, added complexity, and reliance on clustering.

**Strengths:**

The method avoids clustering or costly combinatorial matching by using dense matrix ops, which is attractive for implementation and throughput.

The module is plug-and-play (no extra learnable parameters), can be applied to pre-trained models or used during training, and is evaluated across several tasks (image classification, segmentation, video, etc.)

**Weaknesses:**

The paper strongly emphasizes being "fully differentiable" as a major advantage, enabling "seamless end-to-end training" and learning optimal merging strategies from scratch. However, the core merging process itself, as described in the methodology, appears to be based on a fixed, pre-defined similarity threshold (tau) and a fixed partitioning strategy. It is not immediately clear what is being learned during end-to-end training if these hyperparameters are static.

**Questions:**

Regarding learning, does the model learn better token representations that are more amenable to merging at a fixed tau? Or is the threshold also learned?

In the result tables (1-3), it is not always clear whether baselines were re-run under identical settings (batch size, precision, device) or whether numbers are taken from original papers. Please explicitly state for each baseline whether you: (a) ran it yourself with the same environment, (b) used reported numbers, or (c) used an implementation with matched measurement settings. To further improve reproducibility, it would be useful to provide scripts or a reproducibility table.

---

> ### Author Response · Authors · 2025-11-14
> **Limitation**
>
> Thank you for the review and kind words. I truly appreciate you recognizing my work.
>
> The optimal similarity threshold ($\tau$) and the specific layers for token merging require manual determination, which is a limitation of MaMe for now. But the similarity threshold ($\tau$ ) in MaMe provides a more intuitive interpretation than ToMe's token reduction ratio ($r$), directly reflecting semantic closeness for merging. Future work aims to address this by exploring methods for automatic determination, such as leveraging reinforcement learning or integrating these parameters as learnable components within the model. For other detailed considerations and discussions, please refer to the revised version of this paper.

---

### Note · Authors · 2025-11-14

**Comment:**

MaMe might not be able to attend ICLR. It's sad. But I still think people will like MaMe in the future. MaMe is the best.

**Withdrawal Confirmation:**

I have read and agree with the venue's withdrawal policy on behalf of myself and my co-authors.